# Molecular Survey of Dirofilaria and Leishmania Species in Dogs from Central Balkan

**DOI:** 10.3390/ani12070911

**Published:** 2022-04-02

**Authors:** Suzana Tasić-Otašević, Sara Savić, Maja Jurhar-Pavlova, Jovana Stefanovska, Marko Stalević, Aleksandra Ignjatović, Marina Ranđelović, Bojan Gajić, Aleksandar Cvetkovikj, Simona Gabrielli

**Affiliations:** 1Medical Faculty, University of Niš, 18000 Niš, Serbia; otasevicsuzana@gmail.com (S.T.-O.); stale1995@gmail.com (M.S.); drsalea@yahoo.com (A.I.); marina87nis@gmail.com (M.R.); 2Public Health Institute Niš, 18000 Niš, Serbia; 3Scientific Veterinary Institute Novi Sad, 21113 Novi Sad, Serbia; sara@niv.ns.ac.rs; 4Institute for Microbiology and Parasitology, Medical Faculty, Ss. Cyril and Methodius University in Skopje, 1000 Skopje, North Macedonia; jurharm@yahoo.com; 5Department of Parasitology and Parasitic Diseases, Faculty of Veterinary Medicine-Skopje, Ss. Cyril and Methodius University in Skopje, 1000 Skopje, North Macedonia; jstefanovska@fvm.ukim.edu.mk (J.S.); acvetkovikj@gmail.com (A.C.); 6College of Agriculture and Veterinary Medicine, UAE University, Al Ain P.O. Box 15551, United Arab Emirates; b.gajic@uaeu.ac.ae; 7Department of Public Health and Infectious Diseases, Sapienza University of Rome, Piazzale Aldo Moro 5, 00185 Rome, Italy

**Keywords:** *Dirofilaria* spp., *Leishmania* spp., Central Balkan, dogs

## Abstract

**Simple Summary:**

Current high mobility of pets in Europe demands a safe management and more detailed knowledge of the geographical distribution and prevalence of major health threatening pathogens. Amongst them are *Dirofilaria immitis* and *Dirofilaria repens*, respectively, the causative agents of cardio-pulmonary and subcutaneous dirofilariosis in dogs and cats. Both *Dirofilaria* species can cause human infections. In addition, leishmaniosis is the only tropical vector-borne disease that has been considered endemic in southern Europe, primarily in the Mediterranean. Results of this survey show that Central Balkan is an endemic region for *Dirofilaria immitis*. Coupled with our detection of the *Leishmania infantum* infection in dogs from North Macedonia and its ability to spread to neighboring countries, results of this study warn us that adequate preventive measures are needed to combat the spread of dirofilariosis and leishmaniosis and preserve the health of both dog and human population of this region.

**Abstract:**

Dirofilariosis and leishmaniosis are severe parasitic diseases in dogs, and their causative agents can also be pathogenic to humans. In this study, we conducted a multicentric survey in the regions of Serbia and North Macedonia with the goal to establish an epidemiological scenario of dirofilariosis and leishmaniosis in the territory of Central Balkan. Using molecular analyses, a total of 535 dogs from Northern Serbia (NS), Southern Serbia (SS) and North Macedonia (NM) were screened for the presence of *Dirofilaria* spp. and *Leishmania* spp. We confirmed that Central Balkan is an endemic region for *Dirofilaria (D.) immitis*, as it was found to be the dominant species in this area, with the highest prevalence of 8.75% in NM, followed by NS (6.68%) and a significantly lower prevalence in SS (1.51%). Two dogs (2.5%) from NM were positive for *Leishmania (L.) infantum* infection. None of the dogs from Serbia tested positive for *Leishmania* spp. High prevalence and dominance of *D. immitis* species, and the rising threat of *L. infantum* spread to the territory of Serbia, suggest that preventive measures are of a great necessity to combat the spread of these vector-borne zoonoses.

## 1. Introduction

Determining geographical distribution of infective agents, their reservoir, and their source in particular region is a prerequisite for creating a proper strategy and effective preventive measures that will preclude the spread of vector-borne zoonoses (VBZ) in Europe. Furthermore, successful disease prevention and management primarily depend on the data obtained by continuous research.

Dirofilariosis and leishmaniosis are severe parasitic diseases in dogs, with a high zoonotic potential. *Dirofilaria (D.) immitis* is commonly known as heartworm because, when found in the heart and large blood vessels, this parasite can cause life-threatening disease in dogs, inducing severe damage to the arteries and right cardiac chambers [1]. On the contrary, *D. repens* causes mild infection of cutaneous/subcutaneous tissue of the animals [2].

In addition, leishmaniosis is the only tropical vector-borne disease that has been considered endemic in southern Europe, primarily in the Mediterranean [3]. High prevalence of asymptomatic patients and wide distribution of natural hosts represent crucial predisposing factors for the occurrence of leishmaniosis, but also provide a possibility of the infection spreading to non-endemic regions in the north of the continent [4]. Consequently, literature data suggest an increase in imported cases of leishmaniosis in non-endemic regions of Europe. *Leishmania (L.) infantum* is a causative agent of visceral and cutaneous form of the infection. Canine leishmaniosis due to *L. infantum* infection is a systemic disease with lymphadenomegaly, skin and eye damage, epistaxis, onychogryphosis, loss of body weight and weakness as the most common clinical manifestations [5].

From epidemiological point of view, the prevalence of VBZ correlates with climate and geographic characteristics of the particular region and, more importantly, the presence of competent vectors. Recently, entomological collection carried out on the territory of the Central Balkan region detected biological vectors for *Dirofilaria* species (*Culex pipiens*, biotype *molestus*, and *Aedes vexans)* and for *Leishmania* sp. (such as *Phlebotomus (P.) papatasi*, *P. perfiliewi*, *P. neglectus* and *P. mascittii*) in Serbia [6]. Competent vectors for *Dirofilaria* spp. microfilariae were also collected on the territory of North Macedonia (unpublished data from the National project of Macedonian Ecological Society).

These results inspired us to conduct a systemic survey in the regions of Serbia and North Macedonia, try and obtain results from non-investigated areas, in order to establish an epidemiological scenario of dirofilariosis and leishmaniosis in the territory of the Central Balkan region.

## 2. Materials and Methods

### 2.1. Dog Population

Until the end of 2019, we successfully collected blood samples of 535 dogs, mainly from the territory of Northern Serbia (NS-389 samples), along with samples from Southern Serbia (SS-66 samples) and North Macedonia (NM-80 samples). Investigation included a total of 535 dogs, from dog shelters as well as privately owned (54.9% male, 45.1% female, aged 6 months to 14 years). Dogs were randomly enrolled in the study during their control, preventive examination whereupon they fulfilled the following criteria: (1) they did not leave the area in which they lived; (2) healthy, without any clinical signs compatible with canine filariosis and canine leishmaniosis; and (3) kept outside. A blood sample of 4–5mL was taken from the cephalic vein of each dog using labelled tubes, on request of the dog owners or shelter responsible person within the regular check-up, where 300–500 μL of blood was placed on filter papers for molecular analyses.

### 2.2. Molecular Analyses

Genomic DNA was isolated from dried blood spots using a commercial kit (Dried Blood Spot DNA Isolation Kit, Norgen Biotek Corporation, Thorold, ON, Canada). The detection of *Dirofilaria* and *Leishmania* species was achieved using the following PCR protocols to detect, respectively, the filariod-cox1 gene (650 bp) [7] and the ITS1 (~300 bp) gene of *Leishmania* spp. [8,9].

To test the specificity of the reactions, DNA extracted from *Dirofilaria repens* specimens or from in vitro culture of *Leishmania infantum* promastigotes, and an equivalent volume of double-distilled water was included in each PCR run as positive and negative controls. PCR products were sequenced, in both directions, using the same primers as for the DNA amplification protocol (BioFab Research, Roma, Italy). Gene sequences were aligned using the ClustalW program and compared with those available in GenBank using the BLASTn tool (http://blast.ncbi.nlm.nih.gov/Blast.cgi, accessed on 20 September 2021).

### 2.3. Statistical Analyses

The data are presented as the count and percentage. The comparison of the frequency of following variables: gender, residence, controlled conditions and presence of ectoparasites was performed by the Chi-square test and the Fisher test. The Kolmogorov–Smirnov test indicated that the age of the dogs was not normally distributed. Therefore, the Mann–Whitney test was performed to compare the age difference between infected and uninfected dogs. The null hypothesis was tested with a significance level of 0.05. Epi Info version 7.2.2.6 (CDC, Atlanta, GA, USA) was used for statistical analyses.

## 3. Results

A total of 38 (7.10%) out of 535 tested dogs were positive for *Dirofilaria* DNA and sequence analysis identified *D. immitis* as the dominant species in the area, found in 6.35% of the samples. The highest prevalence of *Dirofilaria* infection, all caused by *D. immitis*, was established in NM (8.75%). Similarly, a high prevalence of *D. immitis* infection in dogs was found in NS (6.68%). *D. repens* was detected in the territory of NS as well, but only in two dogs. The lowest prevalence of the infection, although with detection of both *D. immitis* and *D. repens*, was found in SS (3.02%), a region where dirofilariosis in dogs has not been recorded previously.

As for the *Leishmania* infection, only two dogs from North Macedonia were positive. Sequence analysis of the amplicons marked *L. infantum* as the culprit. Regardless of the considerable number of examined dogs from Serbia, none of them tested positive for *Leishmania* spp. Sampling locations and the detailed distribution of the infected dogs are shown in Figure 1.

Statistical analyses show that infected dogs were significantly older than uninfected ones (*p* = 0.037), with female dogs being infected more frequently (Χ^2^ = 5.3, df = 1, *p* = 0.022). Surprising fact is that urban or rural residence (*p* = 0.162) controlled conditions—accommodation in asylums, regular nutrition, regular hygiene, regular veterinary health care (*p* = 0.617) and the presence of ectoparasites (*p* = 0.561)—are not associated with the occurrence of the infection.

## 4. Discussion

As forecast models predict, epidemiological and epizootiological scenarios of vector-borne diseases constantly change [1]. Climate changes, economic problems, migration of people and animals and changes in vector ecology significantly influence the need for constant monitoring of zoonotic pathogens. The territory of Europe is endemic for dirofilariosis in dogs, with the main distribution in the Mediterranean region, where in several areas both *D. repens* and *D. immitis* coexist [10].

In Serbia, the first systemic research of dirofilariosis in dogs started in the beginning of the 21st century. This investigation demonstrated that the northern part of Serbia (Vojvodina Province) was hyperendemic and endemic for *D. repens* and *D. immitis* infection in dogs, respectively [11,12]. In the same period, studies from Southeastern Serbia, previously a hyperendemic region for *D. repens* infection, found that some areas are still endemic (with significantly lower prevalence of *D. repens* infection detected only in dogs), but declared certain parts as *Dirofilaria*-free zones [13]. In the following years, *Dirofilaria* infection was detected in different hosts, such as red foxes and golden jackals, and present competent biological vectors [14,15,16]. Although North Macedonia shows epidemiological features for dirofilariosis onset, data on *Dirofilaria* infection from this country are scant, not easily found in the international literature and mainly performed using traditional techniques [17]. Cases of dirofilariosis caused by *D. repens* (21%) and *D. immitis* (12.5%) have been reported in few dogs [17], followed by new data of recently reported sporadic *D. repens* infection in wolves [18].

Recently, in addition to dirofilariosis, leishmaniosis caused by *L. infantum* has been diagnosed repeatedly among dogs and humans of the Balkan Peninsula countries, after decades with sporadically reported cases [19].

Since 1999, the number of reported cases of canine and human visceral leishmaniosis in Romania started to rise, with confirmation of autochthonous disease at the same time [20,21]. Leishmaniosis was diagnosed in dogs and humans in Bulgaria as well [22,23,24]. Several studies report positive results in a serological survey of dogs, mostly in rural areas of Albania [25,26]. As for Montenegro and Croatia, it is well known that the Adriatic coast is endemic for leishmaniosis, which was proven in the last decade [27,28,29]. In addition, *Phlebotomus neglectus*, a confirmed vector for *L. infantum*, was identified in the coastal regions of Croatia and Montenegro [19]. The first molecular and serological survey in Bosnia and Herzegovina showed exposure and infection by *L. infantum* in a high percentage of examined dogs [30]. Greece is the only country on the Balkan Peninsula considered endemic for both forms of the disease, with the visceral form being more frequent. Seropositive dogs were found in nearly all surveyed prefectures, with seropositivity as high as 50.2% in the island of Corfu [31,32]. Additionally, *Leishmania* spp. has been detected in stray cats in insular and continental parts of Greece [33,34].

Considering the findings in the surrounding countries, Serbia, as a former endemic area of visceral leishmaniosis, is again at risk of the infection spread. After many years of negligence, this vector-borne disease was found in the territory of Northern Serbia. *L. infantum* was detected in vectors [6] and among the golden jackals (*Canis aureus*) [35]. Moreover, there are reports of dogs positive for anti-*L. infantum* antibodies and among them were dogs that have never left the country [36,37].

Using molecular analyses, we investigated the prevalence of *Dirofilaria* and *Leishmania* infection in asymptomatic dogs from the territory of the Central Balkan region. Besides dogs from Northern Serbia, the study included dogs from Southern Serbia, the territory which previously conducted investigation point as *Dirofilaria*-free zone, and from North Macedonia, from where we have insufficient data.

Results from this study show the highest prevalence of asymptomatic *Dirofilaria* infection (8.75%) in North Macedonia, all due to *D. immitis*. Similarly, 7.45% of infected dogs were found in Northern Serbia, with heartworm being the primary culprit as well. The region of Southern Serbia had the lowest prevalence of the infected dogs, although with both *D. immitis* and *D. repens* species being detected. The whole territory of the Central Balkan region is endemic for dirofilariosis, but the interesting fact is that the epizootiological scenario has changed. Now *D. immitis* dominates in the area. An overall prevalence of about 8.8% for *D. immitis* was shown, suggesting a noticeable risk of heartworm infections in the study region. Our results are different from the findings in recently conducted survey in nearby Romania, where molecular analyses showed that among all filaroids detected in the examined dogs, *D. repens* is the dominant infective agent [38].

In our survey, none of the 455 dog blood samples collected in the territory of Serbia tested positive for *L. infantum*, and this parasite was identified in only two samples from North Macedonia, with a prevalence of 2.5%. Considering that *L. infantum* was previously detected in competent vectors [6], among the jackals [35] and in the human population of the region [39], as well as the reports of seropositive dogs from Northern Serbia [36,37], our findings are unexpected and certain considerations about the limits of this study should be discussed. Despite the blood sample being a reliable clinical material for detecting *Leishmania* DNA from asymptomatic hosts, it showed extremely low levels of parasitemia compared to bone marrow aspirate. To overcome this critical issue, an RT-PCR-based approach is suggested instead of a conventional PCR protocol to detect the parasite in peripheral blood, due to the ability of qPCR to quantify extremely low levels of parasitemia [40]. Thus, the low prevalence of *L. infantum* detected in this study can be attributed to the limited sensitivity of the detection method performed—conventional PCR from dried blood samples, which cannot detect low parasite load.

Statistical analyses of the data show that the infection was significantly more common in older dogs, which is understandable, given the fact that they are exposed to potential sources of the infection for a longer period of time. Infection was also significantly more frequent in female dogs. As there is no relevant data on that matter in the available literature, it could be considered an incidental finding and a result of dog gender not being one of the inclusion criteria for this study. Other than that, we did not determine other significant risk factors in the investigated dogs.

## 5. Conclusions

To conclude, we can emphasize that Central Balkan is an endemic region for *Dirofilaria immitis*, and our findings show that we can expect a significant impact on dog health going forward. Moreover, the domination of *D. immitis* in the study area allows the assumption that more frequent visceral forms of human dirofilariosis may be anticipated as well. Even though we did not determine *Leishmania* infection in dogs from Serbia, findings of *L. infantum* in dogs from North Macedonia show that it can be a threat to the territory of Serbia in the foreseeable future, suggesting that preventive measures are of great necessity to combat the spread of this infection.

## Figures and Tables

**Figure 1 animals-12-00911-f001:**
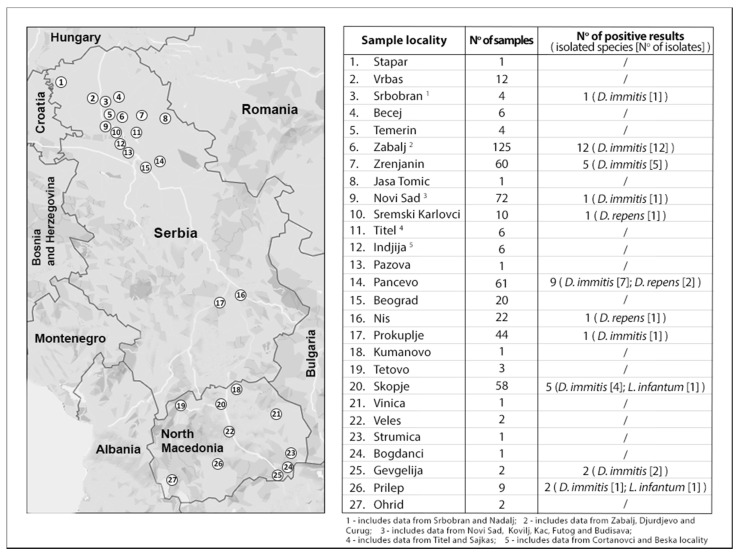
Prevalence of the infection in dogs from different areas of Central Balkan.

## Data Availability

The data presented in the study are available on request from the corresponding author.

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
