# Peer review of "Molecular Survey of Dirofilaria and Leishmania Species in Dogs from Central Balkan"

_animals, 2022, doi:10.3390/ani12070911_

Round 1

Reviewer 1 Report

  1. The title is not suitable, it is not comprehensive.
  2. In 2.3, I could not understand which gene you used in the present study, cox1or ITS1?
  3. The results are simple, you could add more, for example, you could describe the prevalence difference of(1) never moved outside the study area; (2) healthy, asymptomatic; (3) kept outdoors.
  4. The genes amplified in the study should be analyze by phylogenetic relations.
  5. A few typo-errors need to correct and re-verify all over the article carefully, such as:Dirofilaria should be italic. There are some minor grammatical mistakes and typographical errors in the MS, which are to be corrected.

Author Response

Dear reviewer,

Thank you for reading our manuscript (animals-1644535) and suggesting improvements. Regarding your remarks, following corrections are now made:

  • Title of the paper is now changed to “Molecular survey of Dirofilaria and Leishmania species in dogs from Central Balkan”.
  • We have made changes to the section “Molecular analyses” to make our testing methodology clearer to the readers.
  • Selection criteria for the study were: (1) dogs that have never left the area in which they lived, i.e., traveling to foreign countries; (2) were healthy at the time, without any clinical signs and symptoms of the infection; (3) were kept outside. Dogs were selected making sure that they satisfy all of the aforementioned criteria; in regards to that, there were no differences between them. We did make some changes to the section “Dog population” to make this clearer.
  • Corrected formatting and grammatical errors.

We hope you deem our revised manuscript suitable for publication.

Kind regards,

Simona Gabrielli

Reviewer 2 Report

Dear author, your manuscript is clear, and the epidemiological survey straightforward. I would avoit using the terminology dirofilariosis and leishmaniosis when speaking about asymptomatic animals. In that case it is "infection" but not the disease. I would clearly separate heartworm disease which is usually sever and sub-cutaneous, which is completey asymptomatic in >99% of cases in dogs. Both dirofilarioses are usually asymptomatic in humans. Don't focus too much on Leishmania when your survey show 0%. Do not put too much emphasize on "one health" or zoonotic disease. The number of human cases, even for Leishmania infantum infection, are very limited.

Could you clearly make the distinction between owned dogs and shelter dogs, how many of each type, and results for each type.

Best regards

Author Response

Dear reviewer,

Authors thank you for your comments. We have made following changes to the manuscript (animals-1644535) with your suggestions in mind:

  • We acknowledge that the term “infection” should have been used in certain cases, so we made changes accordingly.
  • Discussion about Leishmania is now briefer and more concise.
  • Mentions of “One health approach” are now omitted from the manuscript.

We expect you’ll find this revised version satisfactory.

Kinds regards,

Simona Gabrielli

Reviewer 3 Report

This manuscript presents the results of a survey of dogs in Serbia and Northern Macedonia to investigate Dirofilaria spp. and Leishamania spp. infections. Study is interesting because it deals with zoonotic parasites and a relatively large group of dogs (n = 535) was tested. Nevertheless, the manuscript needs improvement in the points described below.

Major points:

  1. Title should be changed because Leishmania spp. was detected in dogs in Bosnia and Herzegovina using PCR. Please see and compare your results with the findings of the following publication in your manuscript: Colella et al., 2019 (Emerg Infect Dis. 2019, 25(2):385-386. doi: 10.3201/eid2502.181481)
  2. I suggest publishing this study as a short communication instead of a regular article, since the results are presented in a very brief manner and phylogenetic analyses are not provided.
  3. Section 2.4. (Statistical analyses): Please correct this section as it contains mistakes and the sentences do not make logical sense.
  4. Lines 161-164: Please refer to these findings in the discussion section.

Minor points:

  1. Section 2.1. (Study design): This section is unnecessary please remove it.
  2. Please italicize the names of the parasite species (e.g. lines: 149-155).
  3. Please also consider comparing your results with the following studies: Potkonjak et al., 2020 (Comp Immunol Microbiol Infect Dis. 2020, 68:101409. doi: 10.1016/j.cimid.2019.101409); Cimpan et  al., 2022 (Comparative Immunology, Microbiology and Infectious Diseases, 2022, 84, 101793. https://doi.org/10.1016/j.cimid.2022.101793)

Author Response

Dear reviewer,

Thank you for reviewing our manuscript (animals-1644535). Considering your comments, we made following changes to the manuscript:

  • Title of the paper is now changed to “Molecular survey of Dirofilaria and Leishmania species in dogs from Central Balkan”.
  • Publications of Colella et al., 2019; Potkonjak et al., 2020; Cimpan et al. 2022; are now considered in our paper and our results are discussed in relation to theirs as well. References are adjusted accordingly.
  • Section “Study design” is removed from the manuscript.
  • Section “Statistical analyses” is revised to be more concise and comprehensible.
  • Technical and formatting changes.

Hopefully, you’ll find our revised manuscript acceptable for publication.

Kind regards,

Simona Gabrielli

Round 2

Reviewer 1 Report

The authors have addressed the comments and suggestions of this reviewer, and the revised manuscript is acceptable for publication.

Author Response

Dear Reviewer,
Thank you for reading our manuscript (animals-1644535). We have made the changes that Editor you suggested and edited the manuscript as Communication.
We hope you deem our revised manuscript suitable for publication.
Kind regards,
Simona Gabrielli

Reviewer 3 Report

The authors made most of the suggested corrections and improved the article. I am satisfied with these changes and recommend publishing the article in its current form.

Author Response

(The authors gave the same response as above.)
